# Aphid BCR4 Structure and Activity Uncover a New Defensin Peptide Superfamily

**DOI:** 10.3390/ijms232012480

**Published:** 2022-10-18

**Authors:** Karine Loth, Nicolas Parisot, Françoise Paquet, Hugo Terrasson, Catherine Sivignon, Isabelle Rahioui, Mélanie Ribeiro Lopes, Karen Gaget, Gabrielle Duport, Agnès F. Delmas, Vincent Aucagne, Abdelaziz Heddi, Federica Calevro, Pedro da Silva

**Affiliations:** 1Centre de Biophysique Moléculaire, CNRS UPR 4301, 45071 Orléans, France; 2UFR Sciences et Techniques, Université d’Orléans, 45071 Orléans, France; 3Univ Lyon, INSA Lyon, INRAE, BF2I, UMR 203, 69621 Villeurbanne, France; 4Univ Lyon, INRAE, INSA Lyon, BF2I, UMR 203, 69621 Villeurbanne, France

**Keywords:** aphid BCR, 3D structure, defensin peptide, bioinsecticidal peptide, symbiosis

## Abstract

Aphids (Hemiptera: Aphidoidea) are among the most detrimental insects for agricultural plants, and their management is a great challenge in agronomical research. A new class of proteins, called Bacteriocyte-specific Cysteine-Rich (BCR) peptides, provides an alternative to chemical insecticides for pest control. BCRs were initially identified in the pea aphid *Acyrthosiphon pisum*. They are small disulfide bond-rich proteins expressed exclusively in aphid bacteriocytes, the insect cells that host intracellular symbiotic bacteria. Here, we show that one of the *A. pisum* BCRs, BCR4, displays prominent insecticidal activity against the pea aphid, impairing insect survival and nymphal growth, providing evidence for its potential use as a new biopesticide. Our comparative genomics and phylogenetic analyses indicate that BCRs are restricted to the aphid lineage. The 3D structure of BCR4 reveals that this peptide belongs to an as-yet-unknown structural class of peptides and defines a new superfamily of defensins.

## 1. Introduction

Insects are among the most important pests of cultured plants and stored products, causing an estimated yearly loss of hundreds of millions of dollars worldwide [1,2,3]. Aphids, in particular, hold a prominent place among insect pests, as they represent up to 26% of the pests found on the main food crops (maize, wheat, potatoes, sugar beet, barley and tomatoes) grown in temperate climates [4,5]. Aphid control strategies rely almost exclusively on chemical treatments, which cause persistent environmental pollution and lead to the emergence of insect resistance [6]. New ecologically friendly solutions are therefore required to control aphids and other phloem-feeders.

Small Disulfide bond-Rich Proteins (DRPs) extracted from plants or arthropods are promising alternative biopesticide molecules [7,8,9]. These naturally occurring molecules display a very broad range of biological activities, mainly related to host-defense processes [10], and show a high structural and chemical diversity. Their exceptional stability is particularly appealing for drug discovery purposes [11]. The rather rigid three-dimensional conformations imposed by their polycyclic architecture confer DRPs with a strong resistance towards in vivo enzymatic degradation. Taken together, these features contribute to establishing DRPs as an emerging lead compound for the development of novel peptide-based drugs and, more recently, as potential biopesticides in agronomical research [7,8,12,13]. For instance, a knottin DRP extracted from pea seeds, PA1b (Pea Albumin 1, subunit b, 37 amino acids (aa), three disulfide bonds) [9,12,14], is toxic to numerous insects, including aphids, cereal weevils, mosquitos and moths [12].

A new class of DRP, called Bacteriocyte-specific Cysteine-Rich (BCR) peptides, has recently been identified in a major crop pest, the pea aphid, *Acyrthosiphon pisum* [15]. Similar to many other crop pest insects that thrive on unbalanced diets, aphids have evolved long-lasting relationships with endosymbiotic bacteria, and almost all aphids are found in association with the γ3-proteobacterium *Buchnera aphidicola* [16,17,18]. This bacterium supplements the host diet with nutrients that are lacking or limited in their habitats, thereby allowing insects to proliferate, causing major economic, social, and health damage. Neither the host nor the endosymbionts can survive independently. *B. aphidicola* is non-culturable, and insects artificially deprived of their endosymbionts (aposymbiotic) cannot survive or reproduce [19]. The maintenance of this association relies on the compartmentalization of endosymbionts in specialized insect cells, called bacteriocytes [20]. BCRs are encoded by seven orphan genes, and are all expressed exclusively in bacteriocytes of both embryonic and adult aphids [15]. This suggests that BCRs may play a role in bacteriocyte homeostasis, presumably in endosymbiont control, as previously described for antimicrobial peptide coleoptericin A in the cereal–weevil endosymbiosis [21]. Consistent with this hypothesis, it has been shown that BCR1, BCR2, BCR3, BCR4, BCR5 and BCR8 exhibit antimicrobial activity or can permeabilize the membrane of *E. coli* cells [22].

Each BCR peptide consists of a secretory signal peptide and a mature peptide (Figure 1) composed of 44–84 amino acids and containing from six cysteine residues in the case of BCR1-2-4-5 and BCR8 to eight in the case of BCR6 [15]. Between BCR1, BCR2, BCR4 and BCR5, the cysteine-rich region is highly divergent, but the six cysteines have almost identical spacing in the predicted proteins. Three of these peptides, BCR1, BCR4 and BCR5, are encoded by genes found within a genomic region of 20 kbp, suggesting that they may have arisen due to a recent tandem gene duplication. No similarity was found between the other BCR family genes [15]. Intriguingly, the pea aphid BCRs show no significant sequence similarity with genes in species outside the aphid lineage.

DRPs have undergone extensive divergent evolution in their sequence structure and function. They are classified based on their secondary structure orientation, cysteine distribution across the sequence, structure similarities in the disulfide bond patterns and precursor gene sequence [10,24]. Based on the sequence analysis, it was assumed that they could have evolved from defensin-type antimicrobial peptides (AMPs), small proteins containing six to eight cysteines, which are universally found in both animals and plants [15,22].

In this study, we focused on BCR4 of *A. pisum*, a benchmark example of a peptide with antimicrobial activities [22]. We discovered that BCR4 has strong insecticidal activity against the pea aphid. Taking advantage of the recent sequencing of several aphid genomes, which enables the study of gene family diversification through comparative and evolutionary analyses [25,26], we conducted a comparative genomic analysis across 22 aphid species for which sequence information was available. This allowed the identification of 76 new BCR sequences, all restricted to aphid species and not related to any known defensins. Finally, we determined the 3D structure of BCR4 and showed that it belongs to a new structural class of disulfide-rich proteins. Overall, the biochemical analyses, the evolutionary history, and the 3D structure of BCR4 give significant insight into the biological and structural properties of BCRs, and provide evidence for the use of this new defensin superfamily as potential new biopesticides.

## 2. Results

### 2.1. Total Synthesis of BCR4

To investigate the biological activity of members of the BCR peptide family and to resolve their 3D structure by NMR, a pure sample of synthetic BCR4 was produced through total chemical synthesis. As standard automated Fmoc/tBu solid-phase peptide synthesis (SPPS) was unsuccessful (Appendix A), we turned to a native chemical ligation (NCL)-based approach [27], relying on the assembly of two medium-sized peptide segments. Using a recently developed methodology [28,29,30,31,32,33], we coupled a 20-amino-acid (aa) N-2-hydroxy-5-nitrobenzylcysteine (N-Hnb-Cys) crypto-thioester with a 30-aa cysteinyl peptide and obtained the full-length reduced form of BCR4 at high purity (Figure 2). Oxidative folding under thermodynamic control was achieved using a standard protocol [34,35], leading to one major compound featuring three disulfide bridges as evidenced by HRMS analysis (Appendix A).

### 2.2. Antimicrobial Activities of BCR4

To assess the antimicrobial activity of BCR4, we tested the effect of various concentrations (ranging from 5 µM to 80 µM) of this peptide on the growth of the Gram-negative bacterium *Escherichia coli* (strain NM522) and the Gram-positive bacterium *Micrococcus luteus*. Consistent with previous observations [22], a slight inhibition of *E. coli* growth was detected at 5 µM BCR4. This antimicrobial activity increased with BCR4 concentration, and we determined a minimal inhibitory concentration (MIC) of 17.0 ± 2.4 µM, for which no bacterial growth was detected. Comparatively, no antimicrobial activity was detected against the Gram-positive bacterium *Micrococcus luteus* (Table 1).

### 2.3. Insect Bioassays

The insecticidal potential of BCR4 was assayed by oral administration of various concentrations of this peptide to the pea aphid and monitoring of survival. For all BCR4 concentrations tested (5–80 µM), aphid mortality was always significantly higher in the BCR4 aphids compared to the control group, demonstrating a specific effect of BCR4 ingestion on aphid survival (Figure 3). This effect was dependent on BCR4 dosage and duration of treatment. The highest concentration tested (80 µM) also had the strongest lethal effect, with a Lethal Time 50% (LT50) of 1.16 days (Table 2) and no surviving aphids after two days of BCR4 treatment (Figure 3).

Even for concentrations as low as 20 µM, aphids, on average, did not survive more than 4 days, and only one-third reached adulthood. Apart from this effect on aphid survival, the most striking phenotypical effect observed with BCR4 treatment was a statistically significant growth inhibition of the surviving aphids for doses higher than 5 µM, with, for instance, a 60% weight reduction 7 days after ingestion of BCR4 at 27.5 µM (Figure 3B).

### 2.4. Phylogeny of the BCR Family

To decipher the molecular evolution of the BCR peptides, we aimed to identify the whole set of proteins homologous to the seven *A. pisum* BCR proteins. Through an exhaustive search of the NCBI nucleotide and protein databases, coupled with specific inspection of the sequenced aphid species available in the AphidBase database [36], we retrieved a total of 76 new BCR sequences across 20 aphid species (Appendix A) of the 22 for which sequences are available in publicly available databases. We also found one additional sequence from *A. pisum*, bringing the total number of BCRs in this insect to eight. Interestingly, all these sequences have been found in members of the Aphidoidea super-family, thus confirming that BCR proteins are restricted to the aphid lineage [15]. The complete set of 83 BCR sequences was used for amino acid sequence alignment and phylogenetic tree reconstruction (Figure 4). Based on these, the BCR sequences can be grouped into four subfamilies, including homologs of the BCR1-2-4-5, BCR3, BCR6 and BCR8 sequences of *A. pisum*, respectively. The majority of subfamilies have six cysteine residues except for the BCR6 subfamily, which has eight cysteines. The spacing between cysteines within the BCR sequences is fully conserved within subfamilies, thus supporting within-family homologies and phylogenetic positioning (Table 3).

Nineteen aphid species have full genome sequences available (*Acyrthosiphon pisum, Aphis craccivora, Aphis glycines, Aphis gossypii, Aulacorthum solani, Cinara cedri, Diuraphis noxia, Eriosoma lanigerum, Macrosiphum rosae, Melanaphis sacchari, Myzus cerasi, Myzus persicae, Pentalonia nigronervosa, Rhopalosiphum maidis, Rhopalosiphum padi, Schizaphis graminum, Sipha flava, Sitobion avenae, Sitobion miscanthi*). This makes it possible to predict the complete set of BCR peptides encoded by those aphid genomes. In this group, we observed a high variability in the number and distribution of BCR peptides among the four subfamilies (Table 3). No BCRs were found in the *E. lanigerum* (Eriosomatinae subfamily) and *S. flava* (Chaitophorinae subfamily) genomes, and only one in *C. cedri* (Lachninae subfamily) (Figure 5). All members of the Aphidinae subfamily appear to have at least two BCR genes, and *M. rosae* has the largest repertoire of BCR sequences, with 13 distinct sequences distributed across the four BCR subfamilies. While members of the BCR3 and BCR8 subfamilies are present in all Aphidinae species included in this study, members of the BCR1-2-4-5 and BCR6 subfamilies appear restricted to the Macrosiphini aphid tribe, suggesting that the genes encoding those BCRs arose from duplication of the former.

### 2.5. BCR4 Solution Structure

We next determined the 3D structure of BCR4 by NMR spectroscopy. The ^1^H NMR and the natural-abundance ^1^H-^15^N sofast-HMQC spectra [39] of the protein showed a good dispersion of the amide chemical shifts, which is indicative of highly structured peptides. Following standard procedures, the analysis of the set of 2D-TOCSY and NOESY spectra allowed a complete assignment of ^1^H chemical shifts. This assignment was facilitated by heteronuclear ^1^H-^15^N and ^1^H-^13^C NMR spectra, particularly in crowded regions of the ^1^H TOCSY and NOESY spectra corresponding to side chains (BRMB entry 34197). The 3D structures were calculated by considering a total of 923 distance restraints, 16 hydrogen bonds, 88 dihedral angles, and three ambiguous disulfide bridges (Table 4).

The 200 water-refined structures of BCR4 possess three disulfide bridges with identical pairings: C17–C34, C21–C32 and C25–C48. Among them, 20 structures were selected that were in agreement with all NMR experimental data and the standard covalent geometry. Coordinates were deposited as PDB entry 7PQW. Analysis of the 20 final structures with PROCHECK-NMR [40] showed that almost all of the residues (96.9%) are in the most favored or allowed regions of the Ramachandran diagram (Table 4).

BCR4 is folded into a compact globular unit consisting of an N-terminal tail (D1-T13) and an α-helix (I14-V24), followed by an antiparallel β-sheet composed of two short β-strands, β_1_ C34-A37 and β_2_ Q43-P46 (Figure 6). The 3D structure is stabilized by three disulfide bridges linking the α-helix to β_1_ (C17-C34), to the loop C25-Y33 (C21-C32) and to the C-terminal tail (C25-C48), respectively.

A comparison of the BCR4 structure with 3D structures deposited in the PDB using the DALI server [41] identified only one structure with a somewhat similar fold, as defined by a low Z score of 2.1. This structure corresponds to a chimera peptide, which is not a natural peptide (pdb code 1WT7, named BUTX-MTX [42]). This structural comparison thus identifies the 3D structure of BCR4 as being a new fold, which is consistent with it being unique to the aphid lineage.

## 3. Discussion

Thus far, most pest control strategies have relied on the use of systemic chemical pesticides, which are increasingly stigmatized because of their persistence in the environment and their toxicity to non-target organisms [6]. Thus, there is a need for new pest management solutions. Small Disulfide-Rich Proteins (DRPs) extracted from plants or arthropods are promising alternative biopesticide molecules [7,8,12]. Genomic plasticity of DRP-encoding sequences is known to foster the adaptability of organisms and to enable the acquisition of new functions [10]. In this work, we successfully produced in sufficient quantity and purity the folded form of BCR4 (Figure 2), a DRP encoded by the pea aphid genome and part of the BCR family of peptides, which was used to explore the insecticidal and structural properties of this protein. We showed that BCR4 strongly interferes with survival and growth of *A. pisum* in a dose-dependent manner (Figure 3). Its range of activity (5–80 µM, Figure 3) is similar to that displayed by PA1b, a promising plant biopesticide [9,12] active on the same insect target [43]. Importantly, from a functional perspective and as previously reported [22], BCR4 has a significant bactericidal effect on *E. coli* (Table 1), a free-living relative of *B. aphidicola*, the obligatory aphid endosymbiont [22]. Based on these results, we propose that the ingestion of exogenous BCR may block nymphal development and induce aphid death by interfering with the population density of this endosymbiont, essential for nymphal growth and survival [19].

From an evolutionary perspective, large-scale homology searches through genomic, transcriptomic and proteomic databases were performed to complete the repertoire of BCR peptides, which to date have been limited to the seven sequences identified in the pea aphid genome. Importantly, the 76 additional sequences we found (bringing the total number of BCR sequences identified to 83) are all encoded by aphid genomes (Figure 4). Phylogenetic analysis showed that aphid BCRs are organized into four subfamilies, including BCR1-2-4-5, BCR3, BCR6 and BCR8 sequences (Figure 3), respectively. BCRs from those four subfamilies all have six or eight cysteine residues, and we observed a very clear intra-group conservation of cysteine topology (e.g., distribution in the sequence and disulfide pairing). However, as for other DRPs, BCRs present high sequence diversity in their inter-cysteine loops, preventing the detection of any firm homology with arthropod defensins at this point [10,24]. While BCR subfamilies had previously been reported [15], we were able to enrich each of them with many new members (Figure 4). We also showed that (i) aphids present varying numbers of BCR-encoding genes, from 1 to 13; and (ii) while BCR3 and BCR8 homologs are widely present in the aphid lineage, genes from the BCR1-2-4-5 and BCR6 subfamilies are restricted to the Macrosiphini tribe, which includes many major agricultural pests. This suggests a complex evolutionary history involving several events of gene duplication and losses with possible functional diversification of the resulting homologs.

The three-dimensional structure of BCR4 was determined by NMR spectroscopy, and subsequent protein fold analysis revealed that BCR4 belongs to an as-yet-unknown structural class of defensin proteins (Figure 6). Defensins are a well-characterized group of DRPs present in all eukaryote genomes [24]. Shaffee’s phylogenetic work from Shaffee et al. [10,24] showed that defensins consist of two independent and convergent superfamilies, each derived from independent evolutionary events: the so-called trans-defensin superfamily is uniquely composed of the CSαβ (cysteine-stabilized α-helix β-sheet) family and the cis-defensin superfamily is composed of the α-, β-, θ-, and big defensin families. In contrast to the CSαβ family (Figure 7), the CXC motif (X denoting any amino acid residue) of BCR4 is in the first β-strand, leading to a different cysteine bonding pattern, i.e., the one corresponding to the β-defensin fold (the second family of defensin well-known as antimicrobial peptides) [10,24]. The fold of BCR4 is therefore a new type of defensin peptide, suggesting that BCRs constitute a new class of antimicrobial cysteine-rich protein. We here hypothesize that the defensins consist of at least three independent and convergent superfamilies: cis-, trans- and BCR defensins. BCR4 is the first member of this new defensin family with the cysteine bonding pattern of β-defensin and the structural motif of the CSαβ family (Figure 7).

BCRs are orphan genes exclusively expressed in aphid bacteriocytes. This gene family has probably evolved in aphid lineages to ensure several functions related to endosymbiosis, including bacteriocyte homeostasis and endosymbiont control. Studies on the cereal weevil *Sitophilus* have shown that the coleoptericin A (ColA) AMP selectively targets endosymbionts within bacteriocytes and impairs bacterial cytokinesis, thereby regulating bacterial cell division and preventing bacterial exit from weevil bacteriocytes [21,44]. Similar results were obtained in the *Rhizobium*-legumes symbiosis, where nodule-specific secreted peptides called NCRs have been shown to target bacteria and induce an irreversible elongation of bacteria, rendering them metabolically active but unable to multiply in vitro outside plant nodules [45]. In the actinorhizal plant *Alnus glutinosa*, nodules express defensin-like peptides, including the Ag5 peptide that targets symbiont vesicles and increases the permeability of vesicle membranes [46]. Small peptides are becoming emerging molecules that target a variety of functions in host–symbiont interaction, including symbiont control and physiology [47]. Such a convergent evolution pattern is especially interesting with respect to its applications.

Overall, the susceptibility of aphids to BCR peptides may lead to the development of effective strategies for controlling such sap sucking pests. The exploration contained in this work may end up in a new protein family targeted to the specific control of aphids, which are some of the most important pests in global agriculture.

## 4. Materials and Methods

### 4.1. Peptide Synthesis

BCR4 was synthesized through native chemical ligation (NCL) of two peptide segments, followed by oxidative folding to form the three disulfide bridges, using previously described protocols [28,29,30,31,32,33]. Briefly, an N-terminal cysteinyl peptide segment (sequence: H-^21^CAVVCNYTSRPCYCVEAAKERDQWFPYCY^50^D-OH) and a C-terminal crypto-thioester segment (sequence: H-^1^DFDPTEFKGPFPTIEICSK^20^Y-(Hnb)C(S*t*Bu)G-NH_2_) were synthesized on a Prelude peptide synthesizer (Protein Technologies, Tucson, AZ, USA) using standard Fmoc/*t*Bu chemistry at a 25 µmol scale and starting from a Tentagel resin equipped with a Rink’s amide linker. The *N*-2-hydroxy-5-nitrobenzyl (Hnb) group was automatically introduced through resin reductive amination. Both peptide segments were purified by C_18_ reverse phase RP-HPLC on a Nucleosil C_18_ column (300 Å, 250 × 10 mm) (Macherey-Nagel, Düren, Germany), maintained at 25 °C. Solvent A was 0.1% trifluoroacetic acid (TFA) in water and B was 0.1% TFA in acetonitrile. A linear gradient from 20% B to 55% B was applied for 30 min at a flow rate of 3 mL/min and yielded the pure reduced form of BCR4 through collection of the peak identified as the target compound. The purified peptides were then chemoselectively coupled under standard NCL conditions (2 mM peptide segments, 50 mM tris-carboxyethylphosphine, 200 mM 4-mercaptophenylacetic acid, 6 M guanidinium chloride, 200 mM phosphate buffer, pH 6.5, 37 °C, 24 h) [28,29] and purified as before to retrieve the pure reduced form of BCR4 (sequence: H-^1^DFDPTEFKGPFPTIEICSKYCAVVCNYTSRPCYCVEAAKERDQWFPYCY^50^D-OH). The purified compound was further analyzed by ESI-HRMS on a Bruker maXisTM 22 ultra-high-resolution Q-TOF mass spectrometer (Bruker Daltonics, Bremen, Germany), in positive mode and a [M+H]^+^ *m*/*z* ratio of 5897.5883 was found (theoretical monoisotopic *m*/*z* calculated for C_266_H_379_N_62_O_79_S_6_: 5897.5869). Subsequently, the pure reduced form of BCR4 was oxidatively folded in vitro (10 µM peptide concentration, 0.1 mM oxidized glutathione, 1 mM glutathione, 1 mM EDTA, 100 mM TRIS, pH 8.5, 20 °C, 48 h) [34,48] before purification to homogeneity by C_18_ RP-HPLC. Analysis of the purified folded product via ESI-HRMS analysis gave results consistent with the formation of three disulfide bridges ([M+H]^+^ *m*/*z* obtained: 5891.5437; theoretical monoisotopic *m*/*z* calculated for C_266_H_373_N_62_O_79_S_6_: 5891.5400).

### 4.2. Antimicrobial Assays

*E. coli* and *M. luteus* were grown in Lysogeny Broth (LB) and Terrific Broth (TB) medium (Sigma-Aldrich, Saint-Louis, MO, USA), respectively. Antimicrobial assays were performed in sterile 96-well plates with a final volume of 100 μL per well, composed of 50 μL of culture and 50 μL of serially diluted peptides (5–80 µM). *E. coli* and *M. luteus* were added at an OD600 of 0.006 and 0.05, respectively. Plates were incubated at 30 °C for 24 h and growth was measured at 600 nm using a Power wave XS-Biotek plate reader (Bioteck Instrument, Colmar, France). The lowest concentration of BCR4 peptide showing complete inhibition was taken as minimal inhibitory concentration (MIC). All analyses were performed in triplicate, with the results expressed as mean ± standard deviation of mean (SEM).

### 4.3. Insects and Insect Assays

The aphid clone used was *A. pisum* LL01, a long-established alfalfa-collected clone containing only the primary endosymbiont *B. aphidicola.* Aphids were maintained on young broad bean plants (*Vicia faba* L. cv. Aguadulce) at 21 °C, with a photoperiod of 16 h light–8 h dark to obtain strictly parthenogenetic aphid matrilines, that were reared and synchronized as previously described [20].

For toxicity analyses, three groups of ten 1st instar nymphs (aged between 0 and 24 h) were collected and placed in *ad hoc* feeding chambers containing an AP3 artificial diet [49] supplemented with different doses (5 µM to 80 µM) of solubilized BCR4 peptide. Toxicity was evaluated by scoring survival daily over the whole nymphal life of the pea aphid (7 days) [50]. Growth was also measured by weighing adult aphids on a Mettler AE163 analytical microbalance (Mettler Toledo, Columbus, OH, USA) at the closest 10 µg, following the protocol described previously [51,52].

Aphid mortality data for all BCR4 concentrations were analyzed separately in a parametric survival analysis with a log-normal fit. Aphid weights were analyzed by ANOVA followed by Tukey–Kramer HSD test for comparing multiple means. All analyses were performed with JMP software version 11 (SAS Institute Cary USA, MacOS version).

### 4.4. Sequence Analysis and Phylogeny

Homologous BCR protein sequences were retrieved using a combination of TBLASTN and BLASTP [53,54] against the aphid genomes available from (i) AphidBase [36], (ii) the NCBI genome database and (iii) the whole NCBI non-redundant protein, nucleotide and EST databases (with *A. pisum* BCR as blast seeds; see Appendix A Appendix A for a complete list of BCR protein sequences used for the phylogenetic analysis). BCR protein sequences were subjected to multiple sequence alignments using the MUSCLE program [55]. Subsequently, a phylogenetic tree was constructed, using the PhyML method [37] implemented in the Seaview software (v5.0.4) [37,56] (LG model with 4 rate classes), and the reliability of each branch was evaluated using the bootstrap method, with 1000 replicates. Poorly supported branches (<50%) were collapsed using TreeCollapseCL4 [38]. Graphical representation and editing of the phylogenetic tree were performed with FigTree (v1.4.3) [57].

### 4.5. NMR Experiments

Prior to NMR analysis, the synthesized BCR4 peptides were dissolved in H_2_O:D_2_O (9:1 ratio) at a concentration of 0.6 mM and pH was adjusted to 4.8. Then, 2D ^1^H NOESY, 2D ^1^H TOCSY, a sofast-HMQC [39] (^15^N natural abundance) and a ^13^C-HSQC (^13^C natural abundance) were performed at 298 K on an Avance III HD BRUKER 700 MHz spectrometer equipped with a cryoprobe. ^1^H chemical shifts were referenced to the water signal (4.77 ppm at 298 K). NMR data were processed using the Topspin software version 3.2^TM^ (Bruker, Billerica, MA, USA) and analyzed with CCPNMR version 2.2.2 [58].

### 4.6. Structure Calculations

Structures were calculated using the Cristallography and NMR System [59,60] through the automatic assignment software ARIA2 version 2.3 [61] with NOE-derived distances, hydrogen bonds (in accordance with the observation of typical long- or medium-distance NOE cross peak network for β-sheets and α-helices respectively—HN/HN, HN/Hα, Hα/Hα), backbone dihedral angle restraints (determined with the DANGLE program [62]) and three ambiguous disulfide bridges. The ARIA2 protocol, with default parameters used, simulated annealing with torsion angle and Cartesian space dynamics. The iterative process was repeated until the assignment of the NOE cross peaks was complete. The last run for BCR4 was performed with 1000 initial structures and 200 structures were refined in water. Twenty structures were selected on the basis of total energies and restraint violation statistics, to represent the structure of BCR4 in solution. The quality of the final structures was evaluated using PROCHECK-NMR [40] and PROMOTIF [63]. The figures were prepared with PYMOL [64].

### 4.7. Structure Relationship of BCR4 Structure

Comparison of the BCR4 3D structure to the PDB database was performed using the DALI server [41].

## Figures and Tables

**Figure 1 ijms-23-12480-f001:**
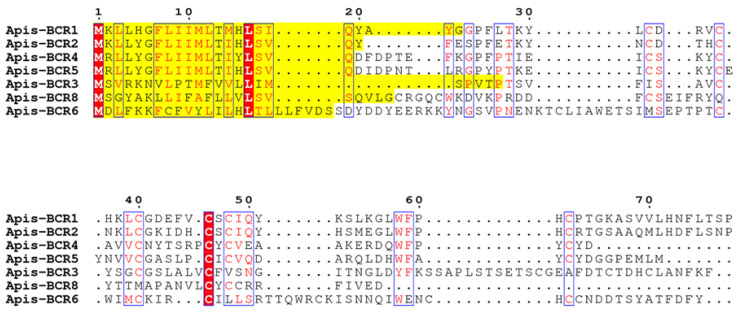
Multiple sequence alignment of *Acyrthosiphon pisum* BCRs. The number and the spacing between these cysteine residues allow to distinguish four BCR sub-families: BCR1-2-4-5, BCR3, BCR8 (containing all six cysteines) and BCR6 (eight cysteines). The signal peptides predicted by SignalP 6.0 server [23] of BCR sequences are highlighted in yellow. The fully conserved residues are framed and highlighted in red.

**Figure 2 ijms-23-12480-f002:**
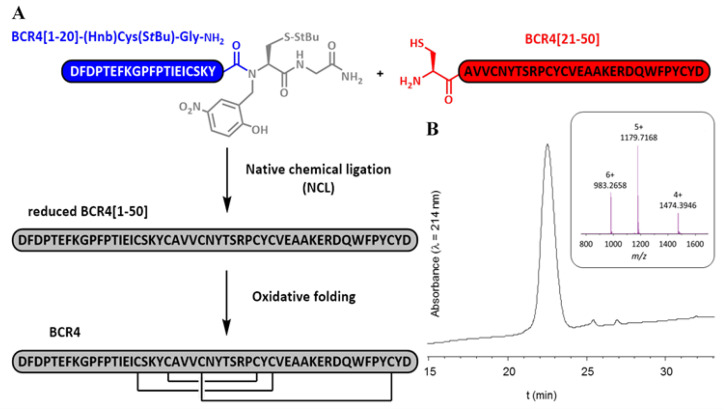
Chemical synthesis of BCR4. (**A**) Schematic representation of the BCR4 peptide chemical through native chemical ligation (NCL) of two peptide segments BCR4[1-20] in blue and BCR4[21-50] in red, followed by oxidative folding to form the three disulfide bridges. (**B**) RP-HPLC chromatogram and ESI-HRMS mass spectrum of the purified folded peptide.

**Figure 3 ijms-23-12480-f003:**
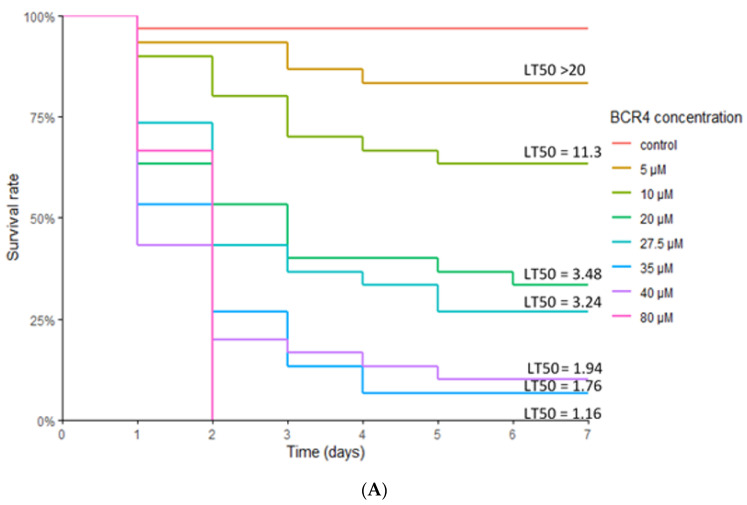
(**A**) Survival curves of the aphid *Acyrthosiphon pisum* reared on artificial diets containing different concentrations of BCR4 peptide. Mean values of Lethal Time 50 (LT50), in days, are indicated above each curve. (**B**) Mass (mg) of 7-day-old pea aphid *Acyrthosiphon pisum* subjected to BCR4 treatment. Concentrations labeled with different letters are significantly different (*p* < 0.05). The surviving aphid sample sizes of BCR4 at 35 and 40 µM were too small to be statistically meaningful.

**Figure 4 ijms-23-12480-f004:**
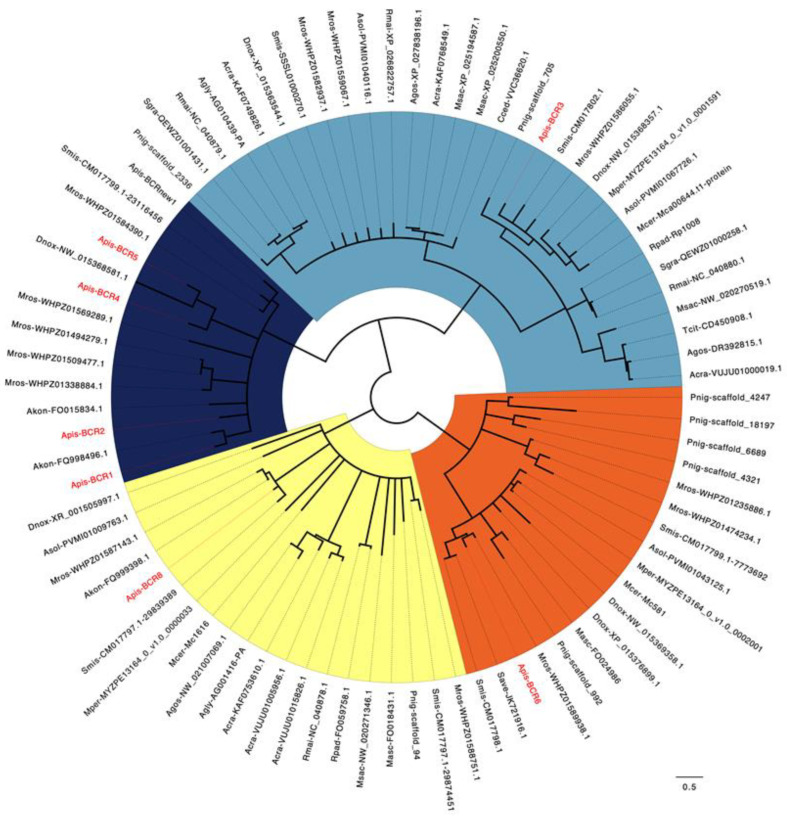
Phylogenetic trees of BCR proteins. Maximum likelihood phylogenetic tree reconstruction was performed using PhyML [37] with an LG 4-rate class model. Branch-support values were calculated using the bootstrap method, with 1000 replicates. Poorly supported branches (<50%) were collapsed using TreeCollapseCL4 [38]. The sequences used for the phylogenetic analysis are listed in Appendix A Appendix A. Sequences labeled in red reflect those identified by the Shigenobu and Stern [15] study in the pea aphid (*A. pisum* genome V3.0). Abbreviations: Acra, *Aphis craccivora*; Agly, *Aphis glycines*; Agos, *Aphis gossypii*; Akon, *Acyrthosiphon kondoi*; Apis, *Acyrthosiphon pisum*; Cced, *Cinara cedri*; Dnox, *Diuraphis noxia*; Dvit. *Daktulosphaira vitifoliae*; Masc, *Myzus ascalonicus*; Mcer, *Myzus cerasi*; Mper, *Myzus persicae*; Msac, *Melanaphis sacchari*; Rmai, *Rhopalosiphum maidis*; Rpad, *Rhopalosiphum padi*; Save, *Sitobion avenae*; Sgra, *Schizaphis graminum*; Tcit, *Toxoptera citricida*.

**Figure 5 ijms-23-12480-f005:**
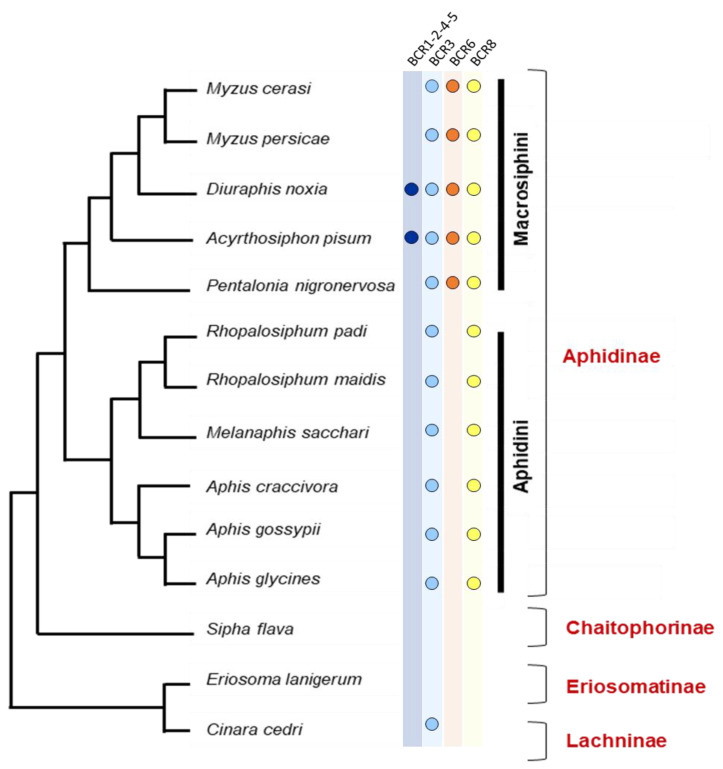
Partitioning of BCR peptides in aphid taxonomic groups. Phylogeny of the 14 members of the aphid group whose genome is annotated and available in databases (among the 19 with sequenced genomes). Their distributions into subfamilies and tribes are indicated in red and black, respectively (adapted from Calevro et al. [4]). Colored dots near each species name indicate whether a member of each BCR subfamily is present in the genome.

**Figure 6 ijms-23-12480-f006:**
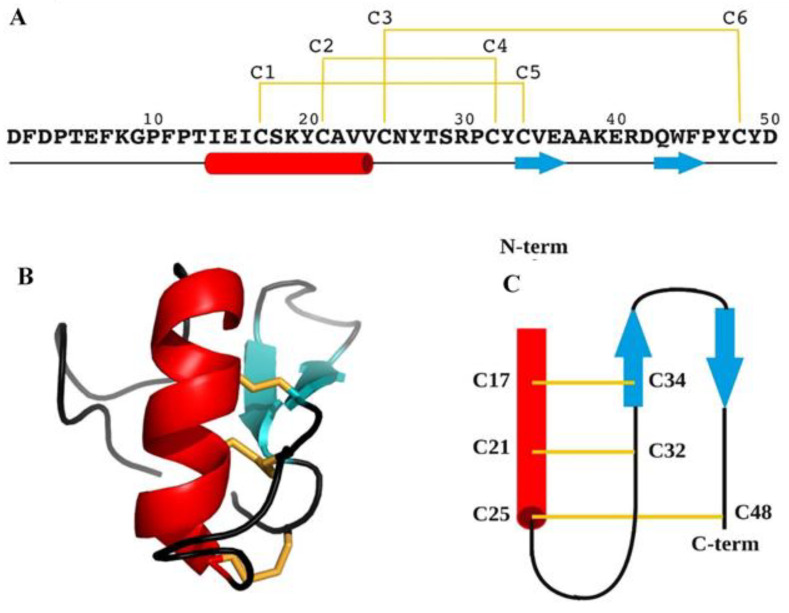
(**A**) Primary structure of BCR4 peptide. (**B**) Three-dimensional structure of BCR4. (**C**) Topology diagram of BCR4. α-helix, β-sheet, and random coil are represented in red, cyan and black, respectively. Disulfide bonds are colored in yellow.

**Figure 7 ijms-23-12480-f007:**
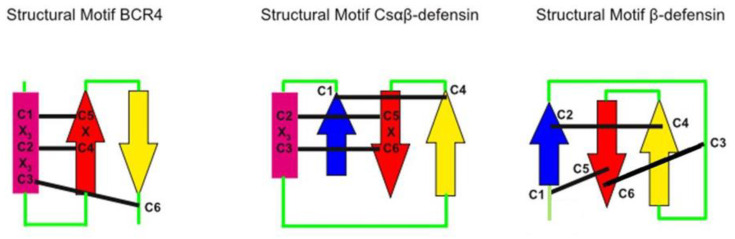
Disulfide connectivities in BCR4 peptide and in CSαβ and β-defensin groups. α-helices are indicated by rectangle, β-strands are represented by arrows, and disulfides in black lines.

**Table 1 ijms-23-12480-t001:** Antimicrobial activities of BCR4 peptide.

	MIC ^a^ (µM)
*Escherichia coli NM522*	17.0 ± 2.4
*Micrococcus luteus*	>80

^a^ Minimal inhibitory concentration.

**Table 2 ijms-23-12480-t002:** Toxicity of BCR4 peptide on the pea aphid *Acyrthosiphon pisum* (mean values of Lethal Time 50 (LT50) in days, with confidence intervals), analyzed by survival analysis with a log-normal fit.

	Concentration (µM)						
**BCR4** **peptide**	80	40	35	27.5	20	10	5
**LT50** **(days)**	1.16 [0.98–1.37]	1.76 [1.27–2.42]	1.94 [1.47–2.57]	3.24 [2.22–4.72]	3.48 [2.28–5.31]	11.3 [4.16–30.8]	>20

**Table 3 ijms-23-12480-t003:** Characteristics of the four BCR subfamilies and of the defensin families.

	Number of Cysteines	Consensus Motif	Cysteine Bonding Pattern	Number of Orthologous Sequences in Aphid Species ^a^
BCR1-2-4-5	6	CX_3_CX_3-5_CX_4-6_CXCX_11-13_C	1:5, 2:4, 3:6	Akon (2), Apis (5), Mros (4), Smis (1), Mros (1), Dnox (1)
BCR3	6	CX_3_CX_6_CX_21-25_CX_6_CX_3_C	Unknown	Acra (3), Agly (1), Agos (2), Apis (1), Asol (2), Cced (1), Dnox (2), Mcer (1), Mper (1), Mros (3), Msac (3), Pnig (2), Rma (3), Rpad (1), Sgra (2), Smis (2), Tcit (1)
BCR6	8	CX_15_CX_3_CX_3_CX_10_CX_10_CXCC	Unknown	Apis (1), Asol (1), Dnox (2), Masc (1), Mcer (1), Mper (1), Mros (3), Pnig (5), Save (1), Smis (2)
BCR8	6	CX_3-4_CX_10_CX_15-17_CXCC	Unknown	Akon (1), Acra (3), Agly (1), Agos (1), Apis (1), Asol (1), Dnox (1), Masc (1), Mcer (1), Mper (1), Msac (1), Mros (2), Pnig (1), Rmai (1), Rpad (1), Smis (2)
CSαβ	6	CX_6-15_CX_3_CX_9-10_CX_4-7_CXC	1:4, 2:5, 3:6	insects
	8	CX_10_CX_5_CX_3_CX_9-10_CX_6-8_CXCX_3_C	1:8, 2:5, 3:6, 4:7	plants
α-defensin	6	CXCX_3-5_CX_9_CX_6-10_CC	1:6, 2:4, 3:5	primates
β-defensin	6	CX_5-7_CX_3-5_CX_8-11_CX_4-6_CC	1:5, 2:4, 3:6	vertebrates

^a^ Abbreviations: Acra, *Aphis craccivora*; Agly, *Aphis glycines*; Agos, *Aphis gossypii*; Apis, *Acyrthosiphon pisum*; Asol, *Aulacorthum solani*; Cced, *Cinara cedri*; Dnox, *Diuraphis noxia*; Mcer, *Myzus cerasi*; Mper, *Myzus persicae*; Msac, *Melanaphis sacchari*; Mros, *Macrosiphum rosae*; Pnig, *Pentalonia nigronervosa*; Rmai, *Rhopalosiphum maidis*; Rpad, *Rhopalosiphum padi*; Save, *Sitobion avenae*; Sgra, *Schizaphis graminum*; Smis, *Sitobion miscanthi*; Tcit, *Toxoptera citricida*. No BCRs were found in *Eriosoma lanigerum *and *Sipha flava*.

**Table 4 ijms-23-12480-t004:** NMR constraints and structural statistics.

NMR Restraints	
*Distance Restraints*	
Total NOE	923
Unambiguous	763
Ambiguous	160
Hydrogen bonds	16
** *Dihedral Angle Restraints* **	88
** *Disulfide bridges ^a^* **	3
**Structural Statistics (7PQW.pdb)**	
** *Average Violations Per Structure* **	
NOEs ≥ 0.3Å	0
Hydrogen bonds ≥ 0.3Å	0
Dihedrals ≥ 15°	3
Dihedrals ≥ 10°	8
Average pairwise rmsd (Å)	backbone atoms 0.45 ± 0.14 heavy atoms 1.14 ± 0.15
**Ramachandran Analysis**	
Most favored region and allowed region	96.9
Generously allowed	0.7
Disallowed	2.4
**Energies (kcal·mol^−1^) ^b^**	
Electrostatic	−1631.48 ± 29.53
Van der Waals	−334.18 ± 11.14
Total energy	−1196.56 ± 31.80
Residual NOE energy	56.38 ± 5.28

^a^ Introduced as ambiguous; ^b^ values are given as mean ± standard deviation (*n* = 20).

## Data Availability

All data are contained within the manuscript and its supporting information.

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
