# Peer review of "Aphid BCR4 Structure and Activity Uncover a New Defensin Peptide Superfamily"

_ijms, 2022, doi:10.3390/ijms232012480_

Round 1
Reviewer 1 Report
Loth and colleagues present an alternative approach to pesticides that utilizes BCR proteins to prohibit the growth of symbiotic bacteria and therefore their aphid hosts. This is especially interesting for agriculture and healthcare (due to the possibility of using a similar approach against parasites and mosquitos), but has broad reaching impact. This paper is well written, well organized, and impactful. There were no major issues noted.
Minor comments:
1) Page 3 line 89: “Tooking advantage” should be “Taking advantage”
2) In Figures S1, S4, and S6, are there any relevant peaks to assign in the crude lysates? Particularly where you would expect to see the BCR4. It would be helpful for all relevant peaks to be assigned in all supplemental figures.
3) Table 1: Have you tried going above 80 µM BCR4 to determine the MIC for Micrococcus luteus? Why stop at 80 µM specifically? What if you used 10x or 100x more?
4) Figure 3: Why is sample size so much lower for 35 and 40 µM?
5) Figure 4: Nice Figure but can you make Figure 4A more readable, even if Figure 4 then takes up 2 pages?
6) What is the mechanism for the antimicrobial activity of the peptide? Since you show structural data, do you think this is largely due to peptide structure? Antimicrobial peptides with less specificity tend to be highly positively charged, enabling them to disrupt negatively charged bacterial cell membranes in general. The BCR4 peptide does not appear to be very positively charged, however: there are actually more negatively charged than positively charged amino acids in BCR4. It would be an interesting addition to the discussion to describe what exactly (or theoretically) about the peptides in your study makes these more specific to the bacteria that are symbiotic with aphids versus other insects.
Author Response
Editing by a native English speaker have been done to improve the manuscript. The changes made are highlighted in the revised version of the manuscript.
Point 1: Page 3, line 89: "Tooking advantage" should be replaced by "Taking advantage".
Response 1: "Tooking advantage" has been corrected to "Taking advantage" page 3 line 94
Point 2: In Figures S1, S4 and S6, are there any relevant peaks to assign in the crude lysates? In particular, where one would expect to see BCR4. It would be helpful if all relevant peaks were assigned in all supplementary figures.
Response 2: Relevant peaks in Figures S1, S4 and S6 have been annotated
Point 3: Table 1: Have you tried going beyond 80 µM BCR4 to determine the MIC for Micrococcus luteus? Why stop specifically at 80 µM? What if you used 10x or 100x more?
Response 3: We did not try further than 80 µM because we were at the limit of peptide solubility.
Point 4: Figure 3: Why is the sample size much smaller for 35 and 40 µM?
Response 4: The sample size is too small for the 35 and 40 µM concentrations because the surviving aphids are too few for statistical analysis. In Figure 3 legend, we have clarified the text as follows “The surviving aphid sample sizes of BCR4 at 35 and 40 µM were too small for statistics” page 5 line 150-151.
Point 5: Figure 4: Nice figure but can you make figure 4A more readable, even if figure 4 then takes 2 pages?
Response 5: To make figure 4 more readable, it has been split into two figures larger. Figure 4A became figure 4 and figure 4B became figure 5. The other figures have been renumbered accordingly.
Point 6: What is the mechanism of the antimicrobial activity of the peptide? Since you are showing structural data, do you think this is largely due to the structure of the peptide? Less specific antimicrobial peptides tend to be strongly positively charged, allowing them to disrupt negatively charged bacterial cell membranes in general. The BCR4 peptide, however, does not appear to be highly positively charged: there are in fact more negatively than positively charged amino acids in BCR4. It would be interesting to add to the discussion a description of the exact (or theoretical) characteristics of the peptides in your study that make them more specific to aphid symbiotic bacteria than to other insects.
Response 6: The mechanism of action of the BCR4 peptide is currently unknown. This work is ongoing. As we have shown in our study, the structure of BCR4 is unique and constitutes a new family of defensins with biophysical properties that may be involved in its specific activity towards aphid symbiotic bacteria. As little is known about this new family, we decided not to discuss further the biophysics properties supposed related to the unknown mechanism of action, even theoretically. The mode of action of BCR4 will certainly be the subject of another nice story.
Reviewer 2 Report
Dear Authors,
the paper IJMS-1958158 can be accepted in the present form, even though there are minor errors to correct, i.e. the Latin names of the cited aphid species have to be written in Italics. Please, look at the attached word revised text of the mns.

Author Response
Point 1: The paper IJMS-1958158 can be accepted in the present form, even though there are minor errors to correct, i.e. the Latin names of the cited aphid species have to be written in Italics.
Response 1: Editing by a native English speaker have been done to improve the manuscript. The changes made are highlighted in the revised version of the manuscript. The Latin names of the cited aphid species have been written in Italics.
Reviewer 3 Report
The manuscript provides a meaningful and excellent contribution to questions pertaining the defensin type of antimicrobial peptides of insects, specifically the small disulfide-rich proteins that are synthesized by bacteriocytes, specific cells that house endosymbiotic bacteria in many insect species. The authors provide convincing evidence that this new class of aphid bacteriocyte-specific cysteine-rich (BCR) provide a new line of investigation for chemical insect pest control.
I have to commend the authors on an organized, well-written and easily comprehensible manuscript. The introductory material is complete, informative and provides in depth information on aphid defensin proteins and small disulfide-rich protein as biopesticides. The author’s use of the literature is very good. To my knowledge this work has not been published elsewhere, the methodology is very well described and straightforward and is appropriate for the questions being investigate. The experimental data is suitable, accurately analyzed and the statistical analyses seem appropriate. The supplemental data is well presented and useful. The conclusions are well stated and supported by their data. The paper is highly appropriate for the journal and the data are scientifically and technically sound, appears repeatable, and are appropriately analyzed.
This paper will be of great interest to all scientist working in insect endosymbiosis and novel pesticides. The overall merit of the paper is significant. The paper is well written with minor grammatical errors. Accept with extremely minor revisions (see below).
Minor revisions:
In the manuscript some species names are in italics and others are not. Please be sure to check the manuscript and in the references for all scientific species names and correct.
One grammatical error on Line 89….Tooking should be Taking.
Otherwise, the manuscript is beautifully written.
Author Response
Point 1: In the manuscript some species names are in italics and others are not. Please be sure to check the manuscript and in the references for all scientific species names and correct.
One grammatical error on Line 89….Tooking should be Taking.
Response 1: Editing by a native English speaker have been done to improve the manuscript. The changes made are highlighted in the revised version of the manuscript. The Latin names of the cited aphid species have been written in Italics. "Tooking advantage" has been corrected to "Taking advantage" page 3 line 94